BioNames: linking taxonomy, texts, and trees

Page Roderic D.M. Roderic.Page@glasgow.ac.uk
Institute of Biodiversity, Animal Health and Comparative Medicine, College of Medical, Veterinary and Life Sciences, Graham Kerr Building, University of Glasgow , Glasgow , UK
Crandall Keith
Electronic publication date: 2013 Oct 29
Publication date: 2013
Volume: 1
Electronic Location ID: e190
Received 2013 Aug 28; Accepted 2013 Oct 8
Copyright: © 2013 Page
Copyright year: 2013
Copyright holder: Page
License: This is an open access article distributed under the terms of the Creative Commons Attribution License, which permits unrestricted use, distribution, and reproduction in any medium, provided the original author and source are credited.
License URL: https://creativecommons.org/licenses/by/3.0/

Keywords: Taxonomy, Database, Nomenclature, Phylogeny, Data integration

Funding: Encyclopedia of Life (EOL) Funding for initial web interface development was provided by the Encyclopedia of Life (EOL). The funder had no role in study design, data collection and analysis, decision to publish, or preparation of the manuscript.

==============================
BioNames is a web database of taxonomic names for animals, linked to the primary literature and, wherever possible, to phylogenetic trees. It aims to provide a taxonomic “dashboard” where at a glance we can see a summary of the taxonomic and phylogenetic information we have for a given taxon and hence provide a quick answer to the basic question “what is this taxon?” BioNames combines classifications from the Global Biodiversity Information Facility (GBIF) and GenBank, images from the Encyclopedia of Life (EOL), animal names from the Index of Organism Names (ION), and bibliographic data from multiple sources including the Biodiversity Heritage Library (BHL) and CrossRef. The user interface includes display of full text articles, interactive timelines of taxonomic publications, and zoomable phylogenies. It is available at http://bionames.org.

Introduction

Large-scale digitisation of biodiversity data is underway on at least three broad fronts. The first, and perhaps the only category that is genuinely “born digital”, is DNA sequencing (Benson et al., 2012). DNA barcoding (Hebert et al., 2003) and, more recently, “metabarcoding” (Taberlet et al., 2012) is generating a flood of sequence data, much of it tied to a specific place and time. The contents of natural history collections are being digitised (Baird, 2010), both the specimens themselves (Blagoderov et al., 2012) and metadata about those specimens. The latter is being aggregated by the Global Biodiversity Information Facility (GBIF; http://data.gbif.org) to provide an overview of the spatial distribution of life on Earth. Much of the biological literature is similarly being converted from physical to digital form, most notably by the Biodiversity Heritage Library (BHL; http://www.biodiversitylibrary.org) (Pilsk et al., 2010). Taxonomic publication is becoming increasingly digital through rise of “mega” journals such as Zootaxa (http://www.mapress.com/zootaxa/) (Zhang, 2006), and semantically enriched journals such as ZooKeys (http://www.pensoft.net/journals/zookeys/) (Penev et al., 2010).

The increasing use of sequence data has made taxonomic relationships readily computable (e.g., by building phylogenetic trees). Yet many DNA sequences are disconnected from classical taxonomy because the associated taxa lack formal taxonomic names (Page, 2011c; Parr et al., 2012). Barcoding has been responsible for a massive influx of these “dark taxa” into the sequence databases (Page, 2011c). Many of these unnamed barcode taxa have since been suppressed by GenBank. But even without the barcoding sequences, dark taxa have been steadily increasing in number in recent years. Names may have a special place in the hearts of taxonomists (Patterson et al., 2010), but the pace of biodiversity discovery is outstripping our ability to put names on taxa, as evidenced by the rise of dark taxa in GenBank. There are increasing calls to adopt less formal taxonomic naming schemes (Schindel & Miller, 2010), or to focus on describing biodiversity without necessarily naming it (Deans, Yoder & Balhoff, 2012; Maddison et al., 2012). A significant challenge will be determining whether these dark taxa represent newly discovered taxa, or come from known taxa but have not been identified as such (Hibbett & Glotzer, 2011; Nagy et al., 2011).

The vision of “Biodiversity Information on Every Desktop” (Edwards, 2000) (perhaps updated to “biodiversity on every device”) rests on our ability to not only digitise life (and the documents we have generated during centuries of cataloguing and studying biodiversity) but also to integrate the wealth of data emerging from sequencing machines and optical scanners. There are numerous points of contact between these different efforts, such as specimen codes, bibliographic identifiers, and GenBank accession numbers (Page, 2008a; Page, 2010). Figure 1 shows a simplified model of the core entities that make up taxonomy and related disciplines (e.g., systematics). The diagram is not meant to be exhaustive, nor does it attempt to rigorously define relationships in terms of one or more available ontologies. Instead, it simply serves as a way to visualise the links between taxon names, the publications (and authors and journals) where those names first appear, the application of those names to taxa, and data associated with those taxa (e.g., DNA sequence-based phylogenies).

Figure 1 Taxonomy data model.

Simplified diagram of the relationships between the core entities that make up taxonomy, such as authors, publications, taxon names, and taxa. Relationships between entities are represented by lines, those in black are the focus of BioNames.

Despite the wealth of possible connections between biodiversity data objects, the most commonly shared identifier that spans sequences, specimens, and publications remains the taxonomic name (Sarkar, 2007; Patterson et al., 2010). We rely on names to integrate data, despite the potential ambiguity in what a given taxonomic name “means” (Kennedy, Kukla & Paterson, 2005; Franz & Cardona-Duque, 2013). Unfortunately, it is often difficult to obtain information on a taxonomic name, either to track its origins and subsequent use, or to verify that it has been correctly used. Typically when taxonomic literature is cited in databases it is as a text string with no link to the growing corpus of digitised literature. Hence taxonomic databases are little more than online collections of 5 × 3 index cards, technology taxonomy’s founding father Linnaeus himself pioneered (Müller-Wille & Charmantier, 2012). Ideally, for any given taxon name we should be able to see the original description, track the fate of that name through successive revisions, and see other related literature. At present this is usually difficult and tedious to do, even in well studied taxa.

EOL Challenge

In response to the Encyclopedia of Life (EOL) Computational Data Challenge (http://eol.org/info/323) I constructed BioNames (http://bionames.org) (Page, 2012). Its goal is to create a database of taxonomic names for animals linked to the primary literature and, wherever possible, to phylogenetic trees. Using existing globally unique identifiers for taxonomic names, concepts, publications, and sequences rather than cryptic text strings (for example, abbreviated bibliographic citations) simplifies the task of linking — we can rely on exact matching of identifiers rather than approximate matching between names for what may or may not be the same entity. This is particularly relevant once we start to aggregate information from different databases, where the same information (e.g., a publication) may be represented by different strings. Furthermore, if we use existing identifiers we increase the potential to connect to other databases (Page, 2008a). This paper outlines how BioNames was built, describes the user interface, and discusses future plans.

Materials & Methods

BioNames integrates data on taxonomic names and classifications, literature, and phylogenies from a variety of sources. Given the inevitable differences in how different databases treat the same data (as well as internal inconsistencies within individual databases), considerable effort must be spent cleaning and reconciling data. Much of this process involves mapping “strings” to “things” (Bollacker et al., 2008), or more precisely, mapping strings to identifiers for things.

Taxon names

At present the taxonomic scope of BioNames is restricted to names covered by the International Code of Zoological Nomenclature (animals and those eukaryotes not covered by the International Code of Nomenclature for algae, fungi, and plants). Taxonomic names were obtained from the Index of Organism Names (ION; http://www.organismnames.com). Each name in ION has a Life Science Identifier (LSID) (Martin, Hohman & Liefeld, 2005) which uniquely identifies that name. LSIDs can be dereferenced to return metadata in Resource Description Framework format (RDF) (Page, 2008b). I used the TDWG LSID resolver (http://lsid.tdwg.org) to obtain the metadata for each LSID. ION LSIDs provide basic information on a taxonomic name using the TDWG Taxon Name LSID Ontology (http://rs.tdwg.org/ontology/voc/TaxonName), in many cases including bibliographic details for the publication where the name first appeared (Fig. 2).

Figure 2 RDF for taxon name.

The RDF retrieved by dereferencing the LSID urn:lsid:organismnames.com:name:371873 which identifies the taxonomic name Pinnotheres atrinicola.

The publication in which the name first appeared is listed in the contents of the “PublishedIn” property. In the example in Fig. 2 this is the string “Description of a new species of Pinnotheres, and redescription of P. novaezelandiae (Brachyura: Pinnotheridae). New Zealand Journal of Zoology, 10(2) 1983: 151–162. 158 (Zoological Record Volume 120)”. I used regular expressions to parse citation strings into their component parts (e.g., article title, journal, volume, pagination), and then attempted to locate the corresponding reference in an external database (see below).

Bibliographic identifiers

When populating BioNames every effort has been made to map each bibliographic string to a corresponding identifier, such as a Digital Object identifier (DOI). For the example in Fig. 2, the citation string “Description of a new species of Pinnotheres, and redescription of P. novaezelandiae (Brachyura: Pinnotheridae). New Zealand Journal of Zoology, 10(2) 1983: 151–162. 158 (Zoological Record Volume 120)” corresponds to the article with the DOI 10.1080/03014223.1983.10423904 (Page, 1983). Once we have a DOI, we can then use services such as those provided by CrossRef (http://www.crossref.org) to retrieve author and publisher information for an article (see Fig. 11 for one use of publisher information).

While DOIs are the best-known bibliographic identifier, there are several others that are relevant to the taxonomic literature (Page, 2009). DOIs are themselves based on Handles (http://hdl.handle.net), an identifier widely used by digital repositories such as DSpace (Smith et al., 2003). A number of journals, such as the Bulletins and Novitates of the American Museum of Natural History, are available in DSpace repositories and consequently have Handles. Other major archives such as JSTOR (http://www.jstor.org/) and the Japanese National Institute of Informatics (CiNii; http://ci.nii.ac.jp/) have their own unique identifiers (typically integer numbers that are part of a URL).

Having a variety of identifiers can complicate the task of finding existing identifiers for a particular publication. Whereas for some identifiers, such as DOIs, CiNii NAIDs (National Institute of Informatics Article IDs), and BioStor reference ids there are search tools (e.g., http://search.crossref.org) or OpenURL resolvers for this task (Van de Sompel & Beit-Arie, 2001) (e.g., http://biostor.org/openurl), for other identifiers there may be no obvious way to find the identifier other than by using a search engine. Another strategy is to build a local database of bibliographic data and match citations strings to that database. I used Mendeley (http://www.mendeley.com) to store bibliographic data harvested from journal or taxon-specific web pages in publicly accessible “groups”, and then queried the local copy of the Mendeley Desktop database to search for references that matched the citation strings.

Identifiers also exist for aggregations of publications, such as journals. The historical practice of abbreviating journal titles in citations has led to a plethora of ways to refer to the same journal. For example, the BioStor database (http://biostor.org; Page, 2011b) has accumulated more than ten variations on the name of the journal Bulletin of Zoological Nomenclature (such as “Bull Zool Nomen”, “Bull Zool Nom.”, ”Bull. Zool. Nomencl.”, etc.). This practice, presumably motivated by the desire to conserve space on the printed page, complicates efforts to match citations to identifiers. One approach to tackling this problem is to map abbreviations to journal-level globally unique identifiers, such as International Standard Serial Numbers (ISSNs) (for the Bulletin of Zoological Nomenclature the ISSN is 0007-5167). In addition to reducing ambiguity, there are web services such as that provided by WorldCat (http://www.worldcat.org) that take ISSNs and return the history of name changes for a journal, which in turn can help clarify the (often complicated) history of long-lived journals. Where possible each journal in BioNames was associated with its corresponding ISSN. If an ISSN is not available for a journal, then the corresponding OCLC Control Number was used as the identifier for the journal.

Documents

Taxonomic publications are available under a variety of licenses, ranging from explicitly open access licenses (MacCallum, 2007) to articles that are “free”, to articles that are behind a paywall. Archives such as JSTOR and CiNii have a mixture of free and subscription-based content. Many smaller journals, often published by scientific societies, are providing their content online for free, if not explicitly under an open license. The Biodiversity Heritage Library (the single largest source of taxonomic articles in BioNames, Fig. 11) makes its content available under a Creative Commons license. Where PDFs were available online either “for free” or under open access, these were downloaded and locally cached. Pages were extracted and converted into bitmap images for subsequent display in a web browser.

Closed-access publications that are available online are linked to by their identifier (e.g., DOI). Access to some of these publications may be available for short-term “rent” by services such as DeepDyve (http://www.deepdyve.com): where possible BioNames includes a link to those services.

Clustering taxonomic names

Taxonomic names comprise a “canonical” name and the name’s authorship, for example Homo sapiens Linnaeus comprises the canonical name “Homo sapiens” and the authorship string “Linnaeus”. Names in taxonomic databases such as ION display numerous variations in spelling of authors and/or variation in the year of publication, and instances of the same canonical name published by different authors (e.g., homonyms), so the names were clustered before populating BioNames. For each set of taxon names with the same canonical name the authorship was compared. If one name lacked an author and the other had an author, the names were automatically merged into a cluster. Given more than two names a graph was constructed where the nodes are the authorship strings, and a pair of nodes is connected if their corresponding strings were sufficiently similar. String similarity was computed by converting the strings to a “fingerprint” comprising lowercase letters with all accented characters replaced by non-accented equivalents, and all punctuation removed, then finding the longest common subsequence of the two strings. By definition the characters in a common subsequence do not need to be consecutive, so the method allows for insertion and deletion of characters. If the length of the subsequence relative to each of the two input strings was longer than a specified threshold (by default, 0.8, where identical strings have a similarity of 1.0) then the two author strings were connected by an edge in the graph. The components of the graph correspond to clusters of names with similar authorship strings, and were treated as being the same name. Figure 3 shows a graph for the different names that all have “Rhacophorus” as the canonical name.

Figure 3 Clustering taxonomic names.

Graph depicting similarity between different authorship strings associated with the name “Rhacophorus”. The components of this graph correspond to the name clusters recognised by BioNames.

Mapping names to taxa

BioNames includes two taxonomic classifications, sourced from GBIF (http://uat.gbif.org/dataset/d7dddbf4-2cf0-4f39-9b2a-bb099caae36c) and NCBI (ftp://ftp.ncbi.nih.gov/pub/taxonomy), respectively. These provide the user with a way to navigate through taxonomic names, as well as view data associated with each classification (e.g., phylogenies). These classifications also provide an explicit definition of the scope of a taxon (i.e., the “taxon concept”). A higher taxon comprises the set of taxa below that taxon in the classification. A terminal taxon (the lowest taxon in a classification) in GBIF can be defined as the set of occurrences linked to that taxon, a terminal taxon in NCBI can be defined as the set of sequences linked to that taxon.

Ideally there would be a one-to-one mapping between a taxonomic name and a taxon, but complications often arise. In addition to the well-known problems of synonymy (more than one name for the same taxon) and homonymy (the same name used for different taxa), name and taxon databases may store slightly different representations of the same name. For example, ION has four records for the name “Nystactes” (each name is followed by its LSID):

Nystactes urn:lsid:organismnames.com:name:2787598

Nystactes Bohlke urn:lsid:organismnames.com:name:2735131

Nystactes Gloger 1827 urn:lsid:organismnames.com:name:4888093

Nystactes Kaup 1829 urn:lsid:organismnames.com:name:4888094

GBIF has three taxa with this name (the number is the GBIF species id):

Nystactes Bohlke, 1957 2403398

Nystactes Gloger, 1827 2475109

Nystactes Kaup, 1829 3239722

Note the differences in the name string (“o” versus “ö” in “Böhlke”, presence or absence of years and commas). To automate the mapping of names to concepts in cases like this I constructed a bipartite graph where the nodes are taxon names, divided into two sets based upon which database they came from (e.g., one set of names from ION, the other from GBIF). I then connect the nodes of the graph by edges whose weights are the similarity of the two strings computed using the longest common subsequence that the two strings share. For example, Fig. 4 shows the graph for “Nystactes”, where the nodes corresponding to ION names are enclosed in ovals, and the names from GBIF are enclosed in rectangles. Computing the maximum weighted bipartite matching of this graph creates a map between the two sets of names. Ideally GBIF should have only one entry for Nystactes because each animal name (with a few exceptions) must be unique. If a newer name has already been published before, then it should be replaced by a new name. In this case, Nystactes (Böhlke, 1957) has since been replaced by Nystactichthys (Böhlke, 1958), and Nystactes Kaup, 1829 (Kaup & Stejneger, 1829) by Paramyotis (Bianchi, 1916). Unfortunately these changes have not yet percolated their way from the primary literature into the GBIF taxonomy.

Figure 4 Matching taxonomic names to taxa.

Bipartite graph of string similarities between taxonomic names containing the string “Nystactes” in the ION and GBIF databases. Solid edges in the graph represent the maximum weighted bipartite matching, and define the mapping between ION name (ovals) and GBIF names (rectangles).

Images

To help the user recognise the taxa being displayed, images for as many taxa as possible were obtained using EOL’s API which provides access to both the images and a mapping between GBIF and NCBI taxon concept identifiers and the corresponding record in EOL.

Phylogenies

Phylogenies were obtained from the PhyLoTA database (http://phylota.net) (Sanderson et al., 2008). This database contains eukaryote phylogenies constructed from automatically assembled clusters of nucleotide sequences (loosely corresponding to “genes”). A MySQL data dump was downloaded (version 184, corresponding to the GenBank release of the same version number) and used to populate a local MySQL database. Metadata for the sequences in each phylogeny was obtained from the European Bioinformatics Institute (EBI; http://www.ebi.ac.uk), and used to populate the MySQL database with basic information such as taxon and locality information, as well as bibliographic details for the sources of the sequences.

Database

Once aggregated, cleaned, and reconciled, the data was converted to JSON (JavaScript Object Notation) and stored in a CouchDB database. CouchDB is a “NoSQL” document database that stores objects in JSON format. Unlike typical SQL databases, CouchDB does not have a database schema and does not support ad hoc queries. Instead CouchDB accepts semi-structured documents, and the developer defines fixed queries or “views” (Anderson, Jan & Noah, 2010).

Results

BioNames comprises a CouchDB database and a web interface. Key features of the interface are outlined below.

Search

BioNames features a simple search interface that takes a scientific name and returns matching taxonomic names and concepts, together with any publications and phylogenies that contain the name. Figure 5 shows an example search result.

Figure 5 Search results.

Screenshot of the search results for a query in BioNames. The results include names that match the query, taxon concepts from GBIF and NCBI with thumbnail images from EOL, phylogenies containing members of the genus, and relevant taxonomic publications.

Document display

BioNames uses the DocumentCloud (https://github.com/documentcloud/document-viewer) viewer to display both PDFs, and page images from digital archives such as BioStor and Gallica (http://gallica.bnf.fr/) (Fig. 6).

Figure 6 Displaying an article.

Screenshot of BioNames displaying a document from BioStor (Conle & Hennemann, 2002). The document viewer can display page images, thumbnails, and (where available) text.

Journals

Much of the work in populating BioNames comprises mapping citation to string to bibliographic identifiers and, where possible, linking those citations to full text. For each journal that has a ISSN, BioNames has a corresponding web page that lists all the articles from that journal that are in the database, and provides a graphical summary of how many of those articles have been located online (Fig. 7).

Figure 7 Displaying a journal.

Screenshot of the page in BioNames for the journal Proceedings of the Entomological Society of Washington (ISSN 0013-8797). The centre column lists the articles in a volume selected by the user using the index on the left. The right hand column displays basic data about the journal, and a graphical display of how many articles have been mapped to a globally unique identifier.

Timeline

BioNames can display timelines of the numbers of taxonomic names published in higher taxonomic groups, inspired by Taxatoy (Sarkar, Schenk & Norton, 2008) (Fig. 8). For a given node in the taxonomic hierarchy the children of that node are displayed as a treemap where the size of each cell is proportional to the log of the number of taxa in the subtree rooted on that child taxon. The number of names in that taxon published in each year is displayed as an interactive chart. Clicking on an individual year will list the corresponding publications for that year.

Figure 8 Timeline of taxonomic names for birds.

Screenshot of the distribution overtime of publications of new names for birds (Aves). The treemap on the left displays taxa below Aves in the taxonomic hierarchy, the chart on the right displays the number of publications in each year that publish a new bird name. The user has clicked on “2012”, resulting in a list of the papers published in that year appearing below the timeline.

Taxa

Each GBIF or NCBI taxon in BioNames has a corresponding web page that lists the associated taxonomic names, publications linked to those names, and other relevant data (e.g., Fig. 9).

Figure 9 Bibliography for a taxon.

Screenshot of the bibliography tab on a taxon page in BioNames. This example shows the publications relevant to the bat genus Rousettus, including those for synonyms. The user can select publications from a given time slice and/or combination of synonyms.

Phylogenies

Phylogenies from PhyLOTA are rendered in an interactive viewer using the Scalable Vector Graphics (SVG) format. The user can zoom in and out, and change the drawing style. Terminal taxa with the same label have the same colour (Fig. 10). This makes it easier to recognise clusters of sequences from the same taxon (e.g., conspecific samples), as well as highlight possible errors (e.g., mislabelled or misidentified sequences). At present the colours are arbitrarily chosen, other schemes could be added in future (Lespinats & Fertil, 2011).

Figure 10 Phylogeny viewer.

Screenshot of phylogeny from PhyLoTA as displayed in BioNames. The user can zoom in and out and pan, as well as change the layout of the tree.

Dashboard

The BioNames web site features a “dashboard” which displays various summaries of the data it contains. For example, Fig. 11 shows a bubble chart of the number of articles different publishers have made available online. “Publisher” in this context is broadly defined to include digital archives such as BioStor and JSTOR, repositories using DSpace, and commercial publishers such as Elsevier, Informa UK, Magnolia Press, Springer, and Wiley.

Figure 11 Relative importance of different publishers of taxonomic literature.

Bubble chart showing relative numbers of taxonomic articles made available online by different publishers.

Discussion

The EOL Computational Data Challenge imposed a deadline on the first release of BioNames, however, development of both the database and web interface is ongoing. Below I discuss some potential applications and future directions.

Links

BioNames makes extensive use of identifiers to clean and link data, but the real value of identifiers becomes apparent when they are shared, that is, when different databases use the same identifiers for the same entities, instead of minting their own. Reusing identifiers can enable unexpected connections between databases. For example, the PubMed biomedical literature database has a record (PMID:948206) for the paper “Monograph on ‘Lithoglyphopsis’ aperta, the snail host of Mekong River Schistosomiasis” (Davis, Kitikoon & Temcharoen, 1976). The PubMed record contains the abstract for the paper, but not a link to where the user can obtain a digital version of the paper. However, this reference is in a volume that has been scanned by the Biodiversity Heritage Library, and the article has been extracted by BioStor (http://biostor.org/reference/102054). If PubMed was linked to BHL, users of PubMed could go straight to the content of the article. But this is just the start. The Davis et al. paper also mentions museum specimens in the collection of the Academy of Natural Sciences of Drexel University, Philadelphia. Metadata for these specimens has been aggregated by GBIF, and the BioStor page for this article displays those links (http://biostor.org/reference/102054). In an ideal world we should be able to seamlessly to traverse the path PubMed → BioStor → GBIF. Likewise, we should be able to traverse the path in the other direction. At present, a user of GBIF simply sees metadata for these specimens and a locality map. They are unaware that these specimens have been cited in a paper (Davis, Kitikoon & Temcharoen, 1976) which demonstrates that the snails host the Mekong River schistosome. This connection would be trivial to make if the reciprocal link was made: GBIF → BioStor. Furthermore, the link BioStor → PubMed would give us access to Medical Subject Headings (MeSH) for the schistosome paper. Hence we could imagine ultimately searching a database of museum specimens (GBIF) using queries from a controlled vocabulary of biomedical terms (MeSH).

Making these connections requires not only that we have digital identifiers, but also that wherever possible, we reuse existing identifiers. In practice forging these links can be hard work (Page, 2011a), and many links may be missing from existing databases (Miller, Norton & Sarkar, 2009). However, if we restrict ourselves to project-specific identifiers then we stymie attempts to create a network of connected biodiversity data.

Text mining

Much of the value of a scientific publication lies dormant unless it is accessible to text mining, which requires access to full text. Where possible BioNames stores information on the publisher of each article (Fig. 11), which could then be used to prioritise discussions with publishers on gaining access to full text (Van Noorden, 2012). Fortunately, the single largest “publisher” of content in BioNames is BioStor (Page, 2011b), which contains scans and OCR text from the Biodiversity Heritage Library. BHL makes its content available under a Creative Commons license, and so can be readily mined. Indeed, the text has already been indexed by tools that can recognise taxonomic names (Akella, Norton & Miller, 2012).

Impact of taxonomic literature

The taxonomic community has long felt disadvantaged by the role of citation-based “impact factor” in assessing the importance of taxonomic research (Garfield, 2001; Krell, 2000; Werner, 2006) especially as much of the taxonomic literature appears in relatively low-impact journals. A common proposal is to include citations to the taxonomic authority for every name mentioned in a scientific paper (Wägele et al., 2011). Regardless of the merits of this idea, in practice these citations are often hard to locate, which is another motivation for BioNames.

There is additional value in surfacing identifiers for the taxonomic literature. In addition to helping construct citation networks, global identifiers can facilitate computing other measures of the value of a taxonomic paper. There is a growing interest in additional measures of post-publication impact of a publication in terms of activity such as social bookmarking, and commentary on web sites (“alt-metrics”) (Yan & Gerstein, 2011). Gathering these metrics is greatly facilitated by using standard bibliographic identifiers (otherwise, how do we know whether two commentators are discussing the same article or not?). If taxonomic literature is to be part of this burgeoning conversation then it needs to be able to be identified unambiguously.

Dark taxa

One of the original motivations for constructing BioNames is the rise of “dark taxa” in genomics databases (Page, 2011c). These are taxa that have been sequenced and added to GenBank, but which lack formal Linnaean names. Typically they will have a name that comprises a genus name and some combination of letters and numbers to make the name unique within GenBank (e.g., a specimen code or the first letter of the lastnames of the researchers that deposited the sequence). It is clear that some dark taxa do, in fact, have names. For example, consider the frog “Gephyromantis aff. blanci MV-2005” (NCBI taxonomy id 321743), which has a single DNA sequence AY848308 associated with it. This sequence was published as part of a DNA barcoding study (Vences et al., 2005). If we enter the accession number AY848308 into Google, we find two documents: one the supplementary table for Vences et al. (2005), the other a subsequent paper (Vences & Riva, 2007) that describes the frog with this sequence as a new species, Gephyromantis runewsweeki. This example is relatively straightforward, but it still required significant time to track down the species description. A key question facing attempts to find names for dark taxa is whether the methods available can be scaled to handle the magnitude of the problem.

Alternatively, one could argue that newer technologies such as DNA barcoding make classical taxonomy less relevant, and perhaps the effort in digitising older literature and exposing the taxonomic names it contains is misplaced. A counter-argument would be that the taxonomic literature potentially contains a wealth of information on ecology, morphology and behaviour, often for taxa in areas that have been subsequently altered by human activity. Given the rarity of many taxa (Lim, Balke & Meier, 2011), and the uneven taxonomic and geographic distribution of taxonomic expertise (May, 1988; Gaston & May, 1992), for many species the only significant data on their biology may reside in the legacy literature (possibly under a different name (Solow, Mound & Gaston, 1995)). As this legacy becomes more accessible through projects such as BHL (and services that build upon that project; Page, 2011a) there will be considerable opportunities to mine that literature for basic biological data (Thessen, Cui & Mozzherin, 2012).

Publishing platform

Recently some taxonomic journals have begun to mark up taxonomic names and descriptions (Penev et al., 2010), which is a precursor to linking names and data together. But these developments leave open the problem of what these links will point to. If we have a database of all taxonomic names and the associated literature (such as BioNames aims to be for zoological names), then such a database would provide an obvious destination for those links. Indeed, ultimately, we could envisage publishing new taxonomic publications within such a database, so that each new publication becomes simply another document within the database (Gerstein & Junker, 2002). In the same way, we could use automated methods to extend the process of tagging names, specimens and literature cited to the legacy literature (Page, 2010), so that the entire body of taxonomic knowledge becomes a single interwoven web of names, citations, publications, and data.

Availability

BioNames is accessible at http://bionames.org. The source code used to build the web site is available on GitHub http://github.com/rdmpage/bionames. Scripts used to fetch, clean, and reconcile the data are archived in http://github.com/rdmpage/bionames-data.

I thank Ryan Schenk for his work on the BioNames, and Cyndy Parr (EOL) for managing the EOL Computational Challenge and providing helpful feedback on the development of BioNames. Mark Holder and an anonymous reviewer provided detailed and helpful comments on the manuscript. Some of the ideas presented here were first explored in a talk at the “Anchoring Biodiversity Information: From Sherborn to the 21st century and beyond” symposium held at The Natural History Museum, London, October 28th 2011. I thank Ellinor Michel for the invitation to speak at that meeting.

Additional Information and Declarations

Competing Interests

Author Contributions

The author declares he has no competing interests.

Roderic D.M. Page conceived and designed the experiments, performed the experiments, analyzed the data, contributed reagents/materials/analysis tools, wrote the paper.

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
