# Peer review of "BioNames: linking taxonomy, texts, and trees"

_PeerJ, doi:10.7717/peerj.190_

## Round 0.1 · original submission · Major Revisions

Rod, I've finally gotten two reviews for your paper. Sorry for the delay, but one reviewer was holding things up for a while. Nevertheless, both reviews are solid and thorough. Both have seemingly legitimate concerns. I think all these concerns are manageable. Both felt the paper was highly relevant and useful and the topic of broad interest. I concur.

Reviewer 1 ·

Basic reporting

The author has sufficiently introduced the topic and described the problems facing taxonomy.

Experimental design

The description of the clustering taxonomic names and the mapping names to taxa (the most significant methodological and not implementation aspect of the paper) was sufficient. The other aspects of the database and how other resources are connected is also discussed in a reasonable manner.

Validity of the findings

The author does demonstrate the there is a functional, though not without errors, database. However, no other findings are presented.

Additional comments

There are a host of things that are very cool about Bionames. First off, it is a necessary project and function that the field desperately needs. There are a number of cool features and tools (new names, journals, generally connecting to many resources, pdf page views, etc.).

I find the fact that general results are not presented very odd. It seems like with all of the information placed into a resource like this, there would be some processing of the data to address some question. This is like publishing a tool announcement without demonstrating that the tool producing interesting results. Of course there are the demonstrations of particular taxa clustering together, presented in the paper, but beyond that there is very little in terms of results. Without this information, there is really very little to review other than some of the issues listed below. I would find it really hard to see this through without the addition of some analyses or summaries of the data. It is hard to judge things like the clustering, for example, without some results.

Should you really call the trees from Phylota phylogenies? They are unrooted (and so have no direction) and so without a fair amount of work are not immediately useful as phylogenies. They are unrooted trees. It would be great to have phylogenies there but calling these unrooted trees phylogenies implies that they contain more information than they actually do.

I would strongly recommend either including more taxonomic scope or changing the name. Bionames only has zoological names but does not state this, as far as I can see, anywhere on the front page. Zoonames maybe?

Along these lines, moving to a larger taxonomic scope will be very problematic because of homonyms. This is not addressed by the author. In this case it seems like the clustering approach will break down and other techniques will need to be used. This seems like a significant barrier and more than just adding another taxonomic dataset. Hopefully the idea would not be to have one of these for each nomenclatural code, most of which, at this point, no longer correspond to clades.

There are some other basic issues, potentially just with implementation that it would seem good to address. I list examples below but the major issue is that clustering taxa into names doesn't seem to be working a good amount of the time (basically all the ones that I tried randomly). Without more general results presented it is hard to tell if there is a basic implementation issue or if this is not the best way to cluster names.

Examples of this issue
Searching Homo and I get three sets two of which seem relavant
http://bionames.org/names/cluster/18997
http://bionames.org/names/cluster/93882
Seems like the clustering isn't working quite right

Pan is also a mess

Lemuridae seems a little better
http://bionames.org/names/cluster/7327
but there is no information on the species. Of course if I go to the NCBI version I get trees, no species, and fairly weak bibliography

·

Basic reporting

Overall, I thought that this contribution was very well written and clear. The context for the contribtution is clear.

There are a few minor points which I think should be revised.
1. abstract "imagery" -> "images"

2. the term 'concept' is used at several spots (e.g line 63), but I don't think that bionames is managing or dealing with the types of definitions of what a taxon "means" in the sense that many folks (e.g. the cited Franz and Cardona-Duque paper) intend when they refer to a taxon concept. It seems important to be clear on what sense you intend when you talk about taxonomic concepts.

3. line 91 "I used" should this be "bionames uses"? If this was a one-time import of references operation, then I have no problem with "I used." But it would be nice to clarify whether bionames periodically updates this citaion info.

4. Bibliographic section. I understand that you are mapping citation strings to DOI's or CiNii NAIDs, and using CrossRef to discover more information about the publication. It is not clear to me what search services are used for the initial citation to identifier search. This seems like an important part of BioNames, so it would be nice to convey what service or series of services are used.

5. Clustering of taxonomic names section (line 141). I think that this needs a tiny bit of clarification. You are only creating edges between nodes representing names from different databases. That is clear from the figure 3 legend (which points out that you're making a bipartite graph). The text in this section should state that. In addition, it might be a bit easier for the reader to understand this if you display the nodes in Figure 3 in two rows (one for names from each source db) or with different symbols (e.g. ovals for ION and rectangles for GBIF).

6. line 154. String subsequence matching. I assume that the string comparison allow for "frameshifts" (which are common because of the existence of multiple ways of turning a glyph with a diacritics into ASCII). Please make that explicit. Also state whether the 80% threshold is 80% of the shorter name or the longer name (when name length varies).


Below are a list of very minor points, that caused me to pause a bit when reading the manuscript. The author may want to consider slight revision of these sections.

1. line 18. It is not clear what you mean by "most notably." Is BHL contributing the most articles, the most important articles, the hardest to obtain via other channels, some combination of these,...?

2. line 20 what makes a journal "mega"

3. line 21. I understand what you mean by "semantically rich" but you might want to add a reference for the uninitiated readers.

4. line 23-24 "DNA sequences are disonnected... they lack formal taxonomic names" reword slightly to clarify that they are not connected to taxa that have formal names (you're not proposing that the sequences themselves, be given taxonomic names).

5. Paragraph starting on line 22. It would be nice to define "dark taxa" here. In the comments your 2011 blogpost, there are clearly some folks who are thrown by the usage of "taxa" for records that correspond to samples. Your intent is pretty clear in this paragraph, but I do think that explicitly stating your definition of the phrase would help.

6. line 53 "Typically taxonomic literature is cited in databases as a text string" would avoid the duplication of "typically" in this sentence.

7. line 58. The claim that obtaining a full history of a name is "almost impossible" seems overstated (given that people do publish monographic revisions of groups). Perhaps "extremely difficult and tedious" ? Or perhaps I'm misunderstanding what it is that you are claiming is "almost impossible"

8. line 84. It would nice to explicily state which web service is used to get the RDF for an ION LSID. It is just the ION metadata link on each taxon's page on ION, right?

9. You mention ISSNs on lines 123-127, but don't explicitly state whether or not BioNames is using ISSN's.

10. "clustering taxonomic names" and Figure 3. It seems to me that seeing a different year would be sufficient to conclude that two names are different (if two names both have dates), but you opted not to use that criterion. It might be helpful to others if you commented on why you avoided that approach.

11. line 189. On the topic of homonyms. GBIF does have a "taxonomicStatus" and a "nomenclaturalStatus" column in its taxonomy. It is probably worth pointing out that all three Nystactes names that it has are listed as "accepted" (taxomonic status) and with no information in the nomenclatural status column. Does BioNames using any of GBIF's info on homonyms?

12. line 209. "the user defines fixed queries or 'views'" The word "user" here is the developer (the user of couchdb, not the user of BioNames), right? You may want to rephrase for "normal" users.

13. line 298. Is there a precedent for this usage of "surfacing" ? Sounds like you're planning on adding a coat of asphalt to the identifiers...

Experimental design

There is no experimental design to comment on. Nor are there findings to report.

This submission is an application note. It is not clear to me whether PeerJ accepts application notes. I must confess some conflict of interest on that question. I have no conflict of interest with this paper, but I can definitely imagine myself submitting a software note to PeerJ if that the journal decides to accept them.

I believe that the normal threshold for whether a discriminating journal will accept a software description is:
1. Is the software non-trivial?
2. Is it potential of interest to other researchers?
3. Is it novel?

I think that the current submission passes these tests with flying colors. Tracking down the connections between names, citations, digitized papers is quite tough and crucial task for many researchers doing basic biology. The author has made this look easy with bionames, but that is only feasible because he has had years of experience working with information technology and this sort of data. I plan on using the tool when I teach systematics and when I conduct my own research. So I think that this description of bionames be a very valuable contribution.

Validity of the findings

See my response to the "Experimental Design" section.

Additional comments

No additional comments.

---

## Round 0.2 · accepted · Accept

Thanks for effective responses to the reviewers and for submitting to PeerJ!